# Sex differences in risk factors for incident peripheral artery disease hospitalisation or death: Cohort study of UK Biobank participants

**Ying Xu**[1,2]*, **Katie Harris**[1], **Anna Louise Pouncey**[3], **Cheryl Carcel**[1], **Gary Low**[1,4,5], **Sanne A. E. Peters**[6,7], **Mark Woodward**[1,7]

1 The George Institute for Global Health, Faculty of Medicine, University of New South Wales, Sydney, New South Wales, Australia, 2 Centre for Health Systems and Safety Research, Australian Institute of Health Innovation, Macquarie University, Sydney, Australia, 3 Department of Vascular Surgery, Division of Surgery and Cancer, Faculty of Medicine, Imperial College London, QEQM, St Mary's Hospital, London, United Kingdom, 4 Research Operations, Nepean Hospital, Nepean Blue Mountain Local Health District, Kingswood, New South Wales, Australia, 5 Sydney Medical School, Faculty of Medicine and Health, University of Sydney, Camperdown, New South Wales, Australia, 6 Julius Center for Health Sciences and Primary Care, University Medical Center Utrecht, Utrecht, The Netherlands, 7 The George Institute for Global Health, School of Public Health, Imperial College London, London, United Kingdom

* yxu1@georgeinstitute.org.au

**Data Availability Statement:** The data that support the findings of this study are available from the UK Biobank, but restrictions apply to the availability of

## Abstract

### Background

Women with peripheral artery disease (PAD) often have atypical symptoms, late hospital presentations, and worse prognosis. Risk factor identification and management are important. We assessed sex differences in associations of risk factors with PAD.

### Methods

500,207 UK Biobank participants (54.5% women, mean age 56.5 years) without prior hospitalisation of PAD at baseline were included. Examined risk factors included blood pressure, smoking, diabetes, lipids, adiposity, history of stroke or myocardial infarction (MI), socioeconomic status, kidney function, C-reactive protein, and alcohol consumption. Poisson and Cox regressions were used to estimate sex-specific incidence of PAD hospitalisation or death, hazard ratios (HRs), and women-to-men ratios of HRs (RHR) with confidence intervals (CIs).

### Results

Over a median of 12.6 years, 2658 women and 5002 men had a documented PAD. Age-adjusted incidence rates were higher in men. Most risk factors were associated with a higher risk of PAD in both sexes. Compared with men, women who were smokers or had a history of stroke or MI had a greater excess risk of PAD (relative to those who never smoked or had no history of stroke or MI): RHR 1.18 (95%CI 1.04, 1.34), 1.26 (1.02, 1.55), and 1.50 (1.25, 1.81), respectively. Higher high-density lipoprotein cholesterol (HDL-C) was more strongly

these data, which were used under license for the current study. Legal constraints do not permit our public sharing of the data. The UK Biobank resources are available from the UK Biobank upon reasonable request and can be accessed through applications on their website (https://www. ukbiobank.ac.uk/).

**Funding:** This work is supported 'GeorgeThink' seed funding from The George Institute for Global Health. ALP is supported by a National Institute for Health and Care Research (NIHR) Doctoral Fellowship. CC is supported by an Australian National Health and Medical Research Council (NHMRC) Investigator Grant, Emerging Leadership 1 (APP2009726). SAEP is supported by a VIDI Fellowship from the Dutch Organization for Health Research and Development (ZonMW) (09150172010050). MW is supported by an Australian NHMRC Investigator Grant, Leadership 2 (APP1174120), and Program Grant (APP1149987). The funders had no role in study design, data collection and analysis, decision to publish, or preparation of the manuscript.

**Competing interests:** : YX, KH, CC, and MW received "GeorgeThink" seed funding from the George Institute for Global Health. ALP is supported by a National Institute for Health and Care Research (NIHR) Doctoral Fellowship. CC is supported by an Australian Heart Foundation Postdoctoral Fellowship (102741) and an Australian National Health and Medical Research Council (NHMRC) Investigator Grant, Emerging Leadership 1 (APP2009726). SAEP is supported by a VIDI Fellowship from the Dutch Organisation for Health Research and Development (ZonMW) (09150172010050). MW is supported by an Australian NHMRC Investigator Grant, Leadership 2 (APP1174120) and Program Grant (APP1149987), and provided recent consultancy for Amgen and Freeline outside the submitted work. The funders had no role in study design, data collection, data analysis, data interpretation, writing of the report, or the decision to submit the paper for publication. All authors have no other competing Interests to declare. These do not alter our adherence to PLOS ONE policies on sharing data and materials.

associated with a lower risk of PAD in women than men, RHR 0.81 (0.68, 0.96). Compared to HDL-C at 40 to 60 mg/dL, the lowest level of HDL-C ($\leq$40 mg/dL) was related to greater excess risk in women, RHR 1.20 (1.02, 1.41), whereas the highest level of HDL-C (>80 mg/dL) was associated with lower risk of PAD in women, but higher risk in men, RHR 0.50 (0.38, 0.65).

## Conclusions

While the incidence of PAD was higher in men, smoking and a history of stroke or MI were more strongly associated with a higher risk of PAD in women than men. HDL-C was more strongly associated with a lower risk of PAD in women than men.

## Introduction

Peripheral arterial disease (PAD) affects more than 236 million people worldwide in 2015 [1]. Men have higher age-standardised cardiovascular disease (CVD) rates [2], and so the significance of CVD, including PAD, in women has been under-recognised and under-investigated [3, 4]. The prevalence and incidence of PAD are similar in women and men with overlapping confidence intervals (CIs), and there is no clear agreement regarding whether women or men exhibit a greater PAD risk [1, 5–8]. There is, however, evidence to suggest sex disparities in PAD presentation, potentially leading to worse prognosis in women. Women with PAD are more often asymptomatic or present with atypical leg symptoms rather than intermittent claudication [9]. They have faster functional decline and greater mobility loss [10]. Women with PAD who underwent vascular surgery have a greater risk of all-cause mortality than men, possibly due to later hospital presentation with chronic limb-threatening ischemia [11].

Although some studies have suggested that sex differences exist in risk factors for PAD, such as smoking, hypertension, diabetes, and C-reactive protein (CRP) [1, 12, 13], the extent of these differences remains to be reliably quantified. Furthermore, an intersectional lens is important, as for instance, the strength of the association between male sex and coronary atherosclerosis was found to vary with ageing [5]. Previous analyses of the UK Biobank cohort have identified several risk factors to be more strongly associated with the risk of myocardial infarction (MI), stroke, and dementia in women compared to men [14–16]. In the current study, we used data from the UK Biobank to examine whether there are sex differences in the associations between major risk factors and incident PAD hospitalisation or death, and we also investigated whether any sex differences varied across major subpopulations.

## Methods

The UK Biobank is a large prospective cohort study of around half a million participants aged 40 to 69 years old recruited between 2006 and 2010. Participants attended one of 22 centres for baseline assessment, where written informed consent was obtained. Touchscreen questionnaires and nurse-led interviews were conducted to collect information on lifestyle and medical history, and physical and functional measurements were taken. The UK Biobank obtained ethics approval from the National Health Service's National Research Ethics Service Committee (ref 21/NW/0157). This research was conducted using UK Biobank resource (application 74018) approved by the access subcommittee.

Sex in the UK Biobank was acquired from central registry at recruitment, but in some cases updated by the participants. Thus, the definition may contain a mixture of the sex the National Health Service recorded for the participant and self-reported sex. In line with our previous work [14–16], we here refer to "sex", "men", and "women". The authors do not had access to information that could identify individual participants.

### Risk factors, covariates, and outcomes

All risk factors and covariates considered in this study were measured at baseline (S1 Table), the majority of which were described in earlier work [14–16]. Participants'baseline data were linked with hospital admission data from England, Scotland, and Wales and the national death register (primary or secondary cause of death) to identify the date of the first record of PAD. Participants who had a hospital admission(s) with PAD before baseline assessment were excluded. Incident PAD documentation, from baseline to 30 September 2021, was the study endpoint. PAD was defined using diagnostic and/or procedure codes (S2 Table). Since some adopted procedures might be conducted for aneurysms, they were only used in the absence of a diagnosis of aneurysms before or at the time of the procedure in a sensitivity analysis. In an additional sensitivity analysis, only diagnostic codes were used. The use of hospital admission data and death register means that the PAD documentation was based on clinical assessments, e.g., clinical history and examination, ankle-brachial index (ABI), duplex ultrasound, or angiography [17], and the disease had caused hospitalisation or death.

### Statistical analysis

Baseline characteristics are presented as number (percentage) for categorical variables and as mean (standard deviation (SD)) or median (the first and third quartiles) for continuous variables as appropriate. Age-adjusted sex-specific incidence of PAD was modelled using Poisson regression. Cox proportional hazard regression models were used to estimate the hazard ratios (HRs) for each risk factor by sex, with interaction terms fitted between each risk factor and sex to obtain the women-to-men ratio of HRs (RHR) [3]. Each risk factor was initially adjusted only for age, then further adjusted for sets of covariates based on perceived probable causal relationships, determined *a priori* [14–16] (Table 1). We assessed the shape of the associations with the risk of PAD for continuous variables, using penalised smoothing splines, adjusted for the same set of covariates as in the Cox models. We conducted subgroup analyses investigating whether sex differences in risk factors for PAD differed by age group (<60 and ≥60 years), by adding an interaction term between sex, age group, and the risk factor of interest to the models. Similarly, subgroup analyses were undertaken by smoking status (never versus ever), diabetes status, and SES (above versus below the UK national median (-0.56)). We also conducted subgroup analyses by antihypertensive medication (use versus not) for blood pressure measures, and by lipid-lowering medication for lipid profiles. In a further sensitivity analysis, we used Fine-Gray regression models and examined the relationships between risk factors and incident PAD documentation considering a competing event, i.e., death due to reasons other than PAD. Participants who did not experience PAD hospitalisation but experienced the competing event were treated as being censored at infinity to indicate that they would never experience PAD [18]. Complete case analyses were performed using R Studio Version 4.0.2 (R Core Team, 2020).

## Results

After excluding 2206 participants (28.3% women, S3 Table) with a previous hospital admission for PAD at baseline, 500,207 individuals (54.5% women, mean age 56.5 years) were included in the analyses. The mean age was 56.3 years (SD 8) for women and 56.7 years (SD 8.2) for

**Table 1. Baseline characteristics of the participants included in main analyses: n(%) unless otherwise stated.**

| Characteristics | Overall (n = 500207) | Women (n = 272704) | Men (n = 227503) |
|---|---|---|---|
| Age (years, mean (SD)) | 56.5 (8.1) | 56.3 (8.0) | 56.7 (8.2) |
| Ethnicity | | | |
| White | 469957 (94.0) | 256562 (94.1) | 213395 (93.8) |
| Other | 30250 (6.0) | 16142 (5.9) | 14108 (6.2) |
| **Blood pressure (mmHg, mean (SD))** | | | |
| Systolic | 137.8 (18.7) | 135.3 (19.2) | 140.9 (17.5) |
| Diastolic | 82.3 (10.2) | 80.7 (10.0) | 84.1 (10.0) |
| Pulse pressure | 55.6 (13.7) | 54.6 (14.3) | 56.8 (12.8) |
| **AHA hypertension categories** | | | |
| Normal | 75128 (15.1) | 55233 (20.3) | 19895 (8.8) |
| Elevated | 61203 (12.3) | 35390 (13.0) | 25813 (11.4) |
| Stage 1 hypertension | 134369 (26.9) | 71983 (26.5) | 62386 (27.5) |
| Stage 2 hypertension | 228196 (45.7) | 109330 (40.2) | 118866 (52.4) |
| Smoking status | | | |
| Never | 273017 (54.9) | 161819 (59.7) | 111198 (49.2) |
| Former | 171874 (34.6) | 85174 (31.4) | 86700 (38.3) |
| Current | 52390 (10.5) | 24207 (8.9) | 28183 (12.5) |
| Smoking intensity (cigarettes per day) | | | |
| $\leq 9$ | 7172 (13.7) | 4392 (18.1) | 2780 (9.9) |
| 10–19 | 15279 (29.2) | 8378 (34.6) | 6901 (24.5) |
| $\geq 20$ | 13274 (25.3) | 5536 (22.9) | 7738 (27.5) |
| Not reported | 16665 (31.8) | 5901 (24.4) | 10764 (38.2) |
| Pack-years among current smokers (mean (SD)) | 28.9 (18.3) | 25.9 (15.8) | 32.1 (20.2) |
| Year since quitting among former smokers (mean (SD)) | 18.7 (11.9) | 18.1 (11.6) | 19.2 (12.1) |
| Type 1 diabetes | 988 (0.2) | 438 (0.2) | 550 (0.2) |
| Type 2 diabetes | 24838 (5.0) | 9825 (3.6) | 15013 (6.6) |
| Cholesterol (mmol/L, mean (SD)) | | | |
| Total cholesterol | 5.7 (1.1) | 5.9 (1.1) | 5.5 (1.1) |
| HDL-C | 1.4 (0.4) | 1.6 (0.4) | 1.3 (0.3) |
| LDL-C | 3.6 (0.9) | 3.6 (0.9) | 3.5 (0.9) |
| Elevated total cholesterol ($\geq 6.2$ mmol/L) | 147563 (31.6) | 93103 (36.7) | 54460 (25.5) |
| HDL-C (mmol/L) categories | | | |
| $\leq 0.78$ | 4212 (1.0) | 508 (0.2) | 3704 (1.9) |
| $>0.78$ and $\leq 1.03$ | 48192 (11.3) | 9571 (4.1) | 38621 (19.6) |
| $>1.03$ and $\leq 1.55$ | 226104 (52.8) | 104920 (45.5) | 121184 (61.5) |
| $>1.55$ and $\leq 2.07$ | 120636 (28.2) | 90746 (39.3) | 29890 (15.2) |
| $>2.07$ | 28825 (6.7) | 25085 (10.9) | 3740 (1.9) |
| BMI (kg/m$^2$, mean (SD)) | 27.4 (4.8) | 27.1 (5.2) | 27.8 (4.2) |
| BMI (kg/m$^2$) categories | | | |
| Underweight ($<18.5$) | 2610 (0.5) | 2073 (0.8) | 537 (0.2) |
| Healthy weight (18.5–24.9) | 161925 (32.6) | 105471 (38.9) | 56454 (25.0) |
| Overweight (25–29.9) | 211216 (42.5) | 99654 (36.7) | 111562 (49.4) |
| Obese ($\geq 30$) | 121415 (24.4) | 64061 (23.6) | 57354 (25.4) |
| Waist circumference (cm, mean (SD)) | 90.3 (13.5) | 84.7 (12.6) | 96.9 (11.3) |
| Waist-to-hip ratio (mean (SD)) | 0.87 (0.09) | 0.82 (0.07) | 0.94 (0.07) |
| Waist-to-height ratio (mean (SD)) | 0.54 (0.08) | 0.52 (0.08) | 0.55 (0.07) |
| History of stroke | 7407 (1.5) | 3090 (1.1) | 4317 (1.9) |

*(Continued)*

**Table 1.** (Continued)

| Characteristics | Overall (n = 500207) | Women (n = 272704) | Men (n = 227503) |
|---|---|---|---|
| History of myocardial infarction | 11109 (2.2) | 2229 (0.8) | 8880 (3.9) |
| Socioeconomic status | | | |
| 1st (least deprived) | 185307 (37.1) | 100838 (37.0) | 84469 (37.2) |
| 2nd | 102378 (20.5) | 56476 (20.7) | 45902 (20.2) |
| 3rd | 74296 (14.9) | 41145 (15.1) | 33151 (14.6) |
| 4th | 66959 (13.4) | 36792 (13.5) | 30167 (13.3) |
| 5th (most deprived) | 70647 (14.1) | 37127 (13.6) | 33520 (14.8) |
| eGFRcys (ml/min/1.73m$^2$, mean (SD)) | 88.3 (16.2) | 88.7 (15.9) | 87.9 (16.4) |
| eGFRcys (ml/min/1.73m$^2$) categories | | | |
| G1 normal or high (≥90) | 226513 (48.5) | 128310 (50.5) | 98203 (46.0) |
| G2 Mildly decreased (60–89) | 219579 (47.0) | 114233 (45.0) | 105346 (49.4) |
| G3a Mildly to moderately decreased (45–59) | 17544 (3.8) | 9608 (3.8) | 7936 (3.7) |
| G3b Moderately to severely decreased (30–44) | 3080 (0.7) | 1535 (0.6) | 1545 (0.7) |
| G4 Severely decreased (15–29) | 629 (0.1) | 266 (0.1) | 363 (0.2) |
| G5 kidney failure (<15) | 116 (0) | 47 (0) | 69 (0) |
| eGFRcys (ml/min/1.73m$^2$) categories | | | |
| Normal or high (≥90) | 226513 (48.5) | 128310 (49.5) | 98203 (46.0) |
| Decreased (<90) | 240948 (51.5) | 125689 (50.5) | 115259 (54.0) |
| C-reactive protein (mg/L, median (Q1, Q3)) | 1.33 (0.66, 2.76) | 1.38 (0.65, 2.97) | 1.28 (0.66, 2.53) |
| Alcohol drinker status | | | |
| Never | 22274 (4.5) | 15908 (5.9) | 6366 (2.8) |
| Previous | 17895 (3.6) | 9909 (3.6) | 7986 (3.5) |
| Current | 458399 (91.9) | 246062 (90.5) | 212337 (93.7) |
| Frequency of alcohol consumption | | | |
| Special occasions only | 57676 (12.6) | 41001 (16.7) | 16675 (7.9) |
| One to three times a month | 55614 (12.1) | 35413 (14.4) | 20201 (9.5) |
| Once or twice a week | 128775 (28.1) | 70044 (28.5) | 58731 (27.7) |
| Three or four times a week | 115045 (25.1) | 55820 (22.7) | 59225 (27.9) |
| Daily or almost daily | 101289 (22.1) | 43784 (17.8) | 57505 (27.1) |
| Use of medication | | | |
| Antihypertensive | 102647 (20.5) | 47566 (17.4) | 55081 (24.2) |
| Lipid-lowering | 85263 (17.0) | 34217 (12.5) | 51046 (22.4) |

AHA denotes American Heart Association, BMI body mass index, eGFRcys estimated Glomerular Filtration Rate calculated using cystatin C, HDL-C high-density lipoprotein cholesterol, LDL-C low-density lipoprotein cholesterol, Q1 the first quartile, Q3 the third quartile, SD standard deviation.

men (Table 1 and S4 Table). Percentages of missing for each variable were <7% (S5 Table), except for high-density lipoprotein cholesterol (HDL-C, 14.4%).

Over 12.6 years (median) of follow-up, 7660 participants (34.7% women) had their first PAD documented (hospitalisation with PAD (n = 7515) or PAD as the cause of death (n = 145)). Overall, the age-adjusted incidence rates of PAD were 7.96 (95%CI 7.56, 8.37) and 18.53 (17.84, 19.21) per 10,000 person-years in women and men, respectively. Male sex was associated with a higher risk of PAD: HR 1.70 (1.61, 1.79) after adjustment for age, systolic blood pressure (SBP), smoking status, diabetes, total cholesterol (TC), body mass index (BMI), history of stroke or MI, SES, estimated Glomerular Filtration Rate calculated using cystatin C (eGFRcys), and lipid lowering and/or antihypertensive medication. The age-adjusted incidence rates of PAD were higher in men than women in categories of all risk factors (Table 2).

**Table 2. Sex-specific age-adjusted incidence of peripheral artery disease (per 10000 person-years) by risk factor status.**

| Risk factor status | Women (n = 272704) | Men (n = 227503) |
|---|---|---|
| | 7.96 (7.56, 8.37) | 18.53 (17.84, 19.21) |
| AHA hypertension categories | | |
| Normal | 4.59 (3.89, 5.29) | 14.80 (12.69, 16.90) |
| Elevated | 6.20 (5.21, 7.20) | 16.09 (14.18, 18.01) |
| Stage 1 hypertension | 6.57 (5.85, 7.29) | 14.83 (13.65, 16.00) |
| Stage 2 hypertension | 11.08 (10.32, 11.83) | 21.57 (20.55, 22.60) |
| Smoking status | | |
| Never | 4.94 (4.52, 5.35) | 8.97 (8.29, 9.65) |
| Former | 9.14 (8.37, 9.92) | 23.76 (22.51, 25.02) |
| Current | 24.47 (22.01, 26.93) | 41.71 (38.68, 44.74) |
| Smoking intensity (cigarettes per day) | | |
| Never smoker | 4.94 (4.52, 5.35) | 8.97 (8.29, 9.65) |
| $\leq 9$ | 16.69 (11.96, 21.41) | 27.40 (19.63, 35.17) |
| 10–19 | 27.80 (23.34, 32.27) | 49.95 (43.21, 56.69) |
| $\geq 20$ | 38.06 (31.53, 44.59) | 62.58 (55.33, 69.84) |
| Diabetes | | |
| No diabetes | 6.94 (6.56, 7.33) | 14.70 (14.07, 15.34) |
| Type 1 | 59.41 (30.12, 88.71) | 85.61 (53.84, 117.39) |
| Type 2 | 32.95 (28.50, 37.40) | 72.21 (66.75, 77.67) |
| Total cholesterol | | |
| Normal (<6.2 mmol/L) | 8.17 (7.63, 8.70) | 20.48 (19.61, 21.34) |
| Elevated ($\geq$6.2 mmol/L) | 7.20 (6.54, 7.87) | 12.01 (10.87, 13.14) |
| HDL-C (mmol/L) | | |
| $\leq 0.78$ | 30.38 (10.84, 49.92) | 52.71 (43.36, 62.05) |
| >0.78 and $\leq$1.03 | 19.95 (16.47, 23.43) | 27.39 (25.35, 29.44) |
| >1.03 and $\leq$1.55 | 8.94 (8.24, 9.63) | 15.71 (14.84, 16.57) |
| >1.55 and $\leq$2.07 | 5.85 (5.25, 6.45) | 12.39 (10.85, 13.94) |
| >2.07 | 5.15 (4.06, 6.23) | 24.96 (18.63, 31.28) |
| BMI (kg/m$^2$) categories | | |
| Underweight (<18.5) | 14.73 (8.21, 21.24) | 37.68 (15.91, 59.46) |
| Healthy weight (18.5–24.9) | 5.74 (5.18, 6.30) | 15.05 (13.81, 16.30) |
| Overweight (25–29.9) | 7.73 (7.07, 8.39) | 15.33 (14.45, 16.22) |
| Obese ($\geq$30) | 11.40 (10.39, 12.41) | 27.68 (25.99, 29.37) |
| History of stroke | | |
| No | 7.65 (7.24, 8.05) | 17.60 (16.93, 18.28) |
| Yes | 37.00 (28.55, 45.45) | 69.03 (59.10, 78.96) |
| History of myocardial infarction | | |
| No | 7.56 (7.17, 7.96) | 16.35 (15.69, 17.01) |
| Yes | 59.45 (46.86, 72.04) | 75.54 (68.28, 82.80) |
| Socioeconomic status | | |
| 1st (least deprived) | 6.08 (5.50, 6.66) | 13.91 (12.94, 14.87) |
| 2nd | 6.76 (5.94, 7.58) | 16.90 (15.44, 18.35) |
| 3rd | 8.33 (7.25, 9.40) | 18.40 (16.60, 20.20) |
| 4th | 9.47 (8.25, 10.68) | 20.30 (18.30, 22.29) |
| 5th (most deprived) | 13.19 (11.75, 14.63) | 31.91 (29.50, 34.32) |
| eGFRcys (ml/min/1.73m$^2$) categories | | |
| Normal or high ($\geq$90) | 3.81 (3.40, 4.22) | 9.15 (8.41, 9.89) |

*(Continued)*

**Table 2.** (Continued)

| Risk factor status | Women (n = 272704) | Men (n = 227503) |
|---|---|---|
| Decreased ($<$90) | 9.67 (8.98, 10.36) | 21.82 (20.73, 22.92) |
| Alcohol drinker status | | |
| Never | 11.82 (9.76, 13.88) | 20.21 (15.87, 24.55) |
| Previous | 17.73 (14.49, 20.97) | 34.85 (29.66, 40.05) |
| Current | 7.30 (6.89, 7.71) | 17.85 (17.15, 18.54) |
| Frequency of alcohol consumption | | |
| Special occasions only | 11.66 (10.38, 12.94) | 26.44 (23.38, 29.51) |
| One to three times a month | 7.65 (6.54, 8.76) | 18.08 (15.79, 20.38) |
| Once or twice a week | 6.16 (5.45, 6.86) | 16.95 (15.65, 18.24) |
| Three or four times a week | 5.22 (4.49, 5.94) | 14 (12.84, 15.17) |
| Daily or almost daily | 7.50 (6.51, 8.48) | 20.23 (18.81, 21.66) |

AHA denotes American Heart Association, BMI body mass index, eGFRcys estimated Glomerular Filtration Rate calculated using cystatin C, HDL-C high-density lipoprotein cholesterol.

## Blood pressure

Higher pulse pressure, and stage 2 hypertension were associated with a higher risk of PAD, similarly in both sexes, whereas higher diastolic blood pressure (DBP) was related to a lower risk of PAD in men only: HR 0.97 (0.95, 0.98) (Fig 1). Per 10 mmHg higher SBP was slightly more strongly associated with the risk of PAD in women compared to men, with a multivariable adjusted women-to-men RHR of 1.02 (1.00, 1.05) (Fig 2). The spline analyses showed a log-linear association between SBP and PAD in women, whereas for men it was J-shaped (log-linear from around 140 mmHg, below which the hazard ratio was mostly flat at one). For DBP, there tended to be U-shaped relationships with the risk of PAD for both sexes (S1 Fig).

## Smoking

For both sexes, compared with those who never smoked, former or current smokers had higher risk of PAD. The HR for former smokers was lower in women 1.61 (1.47, 1.76) than in men 2.00 (1.86, 2.14), and the women-to-men RHR was 0.81 (0.72, 0.90). Conversely, compared with never smoking, the HR for current smoking was greater in women than in men, women-to-men RHR of 1.18 (1.04, 1.34). There was a sex difference in the HRs at the greatest smoking intensity of $\geq$20 cigarettes per day, compared to never smoking, where the excess risk was greater for women than men, RHR 1.21 (1.01, 1.45).

## Diabetes

Compared to those without diabetes, type 1 diabetes was related to a higher risk of PAD, HR 5.82 (3.87, 8.77) in women and 4.56 (3.39, 6.14) in men. For type 2 diabetes, the HR was 1.99 (1.75, 2.28) in women and 2.28 (2.11, 2.47) in men. There was no evidence for sex differences.

## Lipids

TC, elevated TC, or low-density lipoprotein cholesterol (LDL-C) were not associated with the risk of PAD in women or men. There was an inverse association between higher levels of HDL-C and the risk of PAD in both sexes, HRs 0.66 (0.58, 0.75) in women and 0.82 (0.73, 0.91) in men with per 1 mmol/L higher HDL-C. The HR was lower among women than men, RHR 0.81 (0.68, 0.96). Compared to HDL-C at 40 to 60 mg/dL, the lowest level of HDL-C

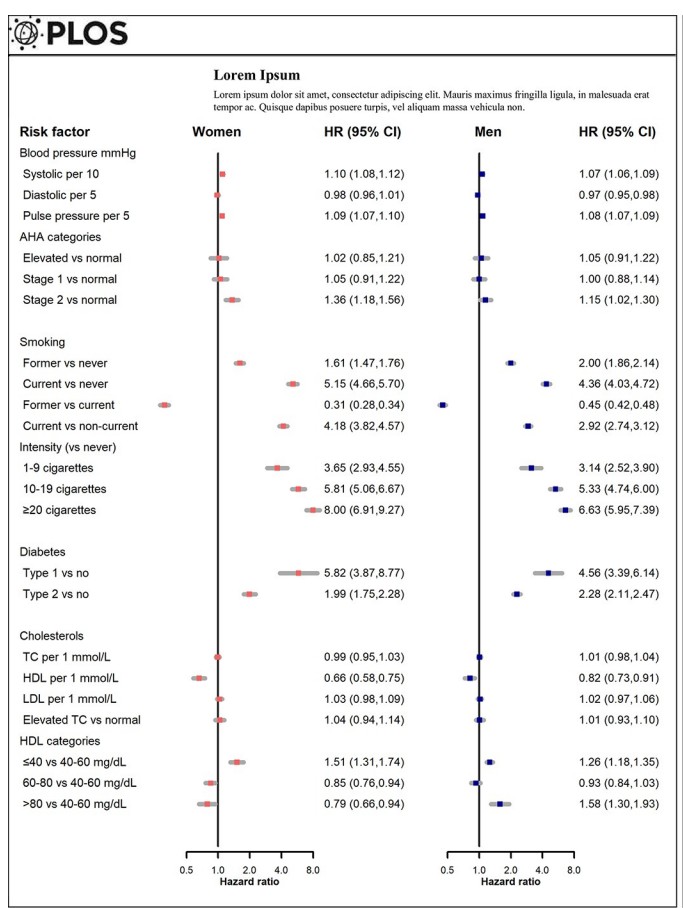

**Fig 1. Sex-specific multivariable-adjusted hazard ratios for traditional risk factors and PAD.** AHA denotes American Heart Association, HDL high-density lipoprotein cholesterol, LDL low-density lipoprotein cholesterol, TC total cholesterol. Pink and blue squares represent hazard ratios for women and men, respectively. Horizontal lines indicate corresponding 95% confidence intervals.

($\leq$40 mg/dL) was related to higher excess risk in women than men RHR 1.20 (1.02, 1.41); whereas the highest level of HDL-C (>80 mg/dL) was associated with lower risk of PAD in women HR 0.79 (0.66, 0.94), but higher risk in men HR 1.58 (1.30, 1.93), women-to-men RHR 0.50 (0.38, 0.65). Higher HDL-C was log-linearly associated with a lower risk of PAD in women, whereas a U-shaped pattern was found in men (S2 Fig).

## Adiposity

Higher BMI, being obese, higher waist circumference, and higher waist-to-hip and waist-to-height ratios were all associated with a higher risk of PAD in both sexes (Fig 3). There are some sex differences (Fig 4). Per 5 kg/m$^2$ higher BMI was associated with 27% higher risk of PAD in women versus 33% higher risk in men, and the women-to-men RHR was 0.95 (0.91, 0.99). HRs associated with overweight compared to healthy weight were 1.18 (1.07, 1.30) in women and 0.98 (0.90, 1.05) in men. The corresponding RHR was 1.21 (1.07, 1.37). Per 0.1 higher waist-to-hip ratio was related to greater excess risk of PAD in men than women, RHR 0.86 (0.83, 0.90). The splines revealed log-linear relationships between BMI, waist circumference, and waist-to-hip and waist-to-height ratios and the risk of PAD in women compared to J-shaped relationships in men (S3 Fig).

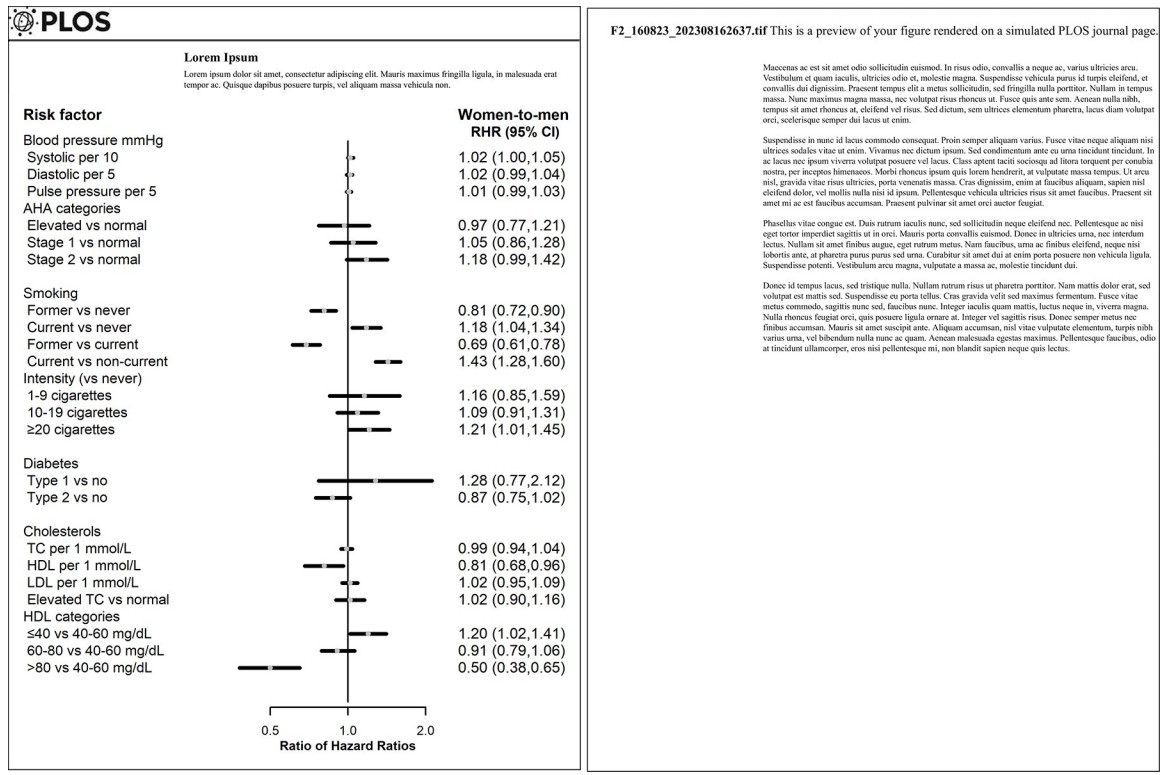

**Fig 2. Women-to-men ratio of hazard ratios for traditional risk factors and PAD.** AHA denotes American Heart Association, HDL high-density lipoprotein cholesterol, LDL low-density lipoprotein cholesterol, TC total cholesterol. Squares represent women-to-men ratio of hazard ratios. Horizontal lines indicate corresponding 95% confidence intervals.

### Prior stroke and MI

The HR for PAD associated with prior stroke was 3.34 (2.81, 3.99) in women and 2.66 (2.38, 2.98) in men. The HR for PAD associated with prior MI was 4.74 (4.01, 5.59) in women and 3.15 (2.91, 3.41) in men. Prior stroke or MI was associated with a greater excess risk of PAD in women than men: corresponding women-to-men RHR 1.26 (1.02, 1.55) for stroke and 1.50 (1.25, 1.81) for MI.

### SES

There were graded log-linear relationships between SES and the risk of PAD in both sexes. A sex difference was seen in the comparison between the most deprived and the least deprived SES groups, where the most deprived group was related to greater excess PAD risk in men: the women-to-men RHR 0.86 (0.74, 0.99). Linear relationships were shown between Townsend scores and PAD in both sexes (S4 Fig).

### Kidney function

Per 10 ml/min/1.73m$^2$ higher eGFRcys was associated with up to 20% lower risk of PAD in both sexes. eGFRcys <90 ml/min/1.73m$^2$ was related to a higher risk of PAD similarly in women HR 1.38 (1.24, 1.53) and men 1.47 (1.37, 1.59). Higher eGFRcys was log-linearly associated with a lower risk of PAD similarly in women and men (S5 Fig).

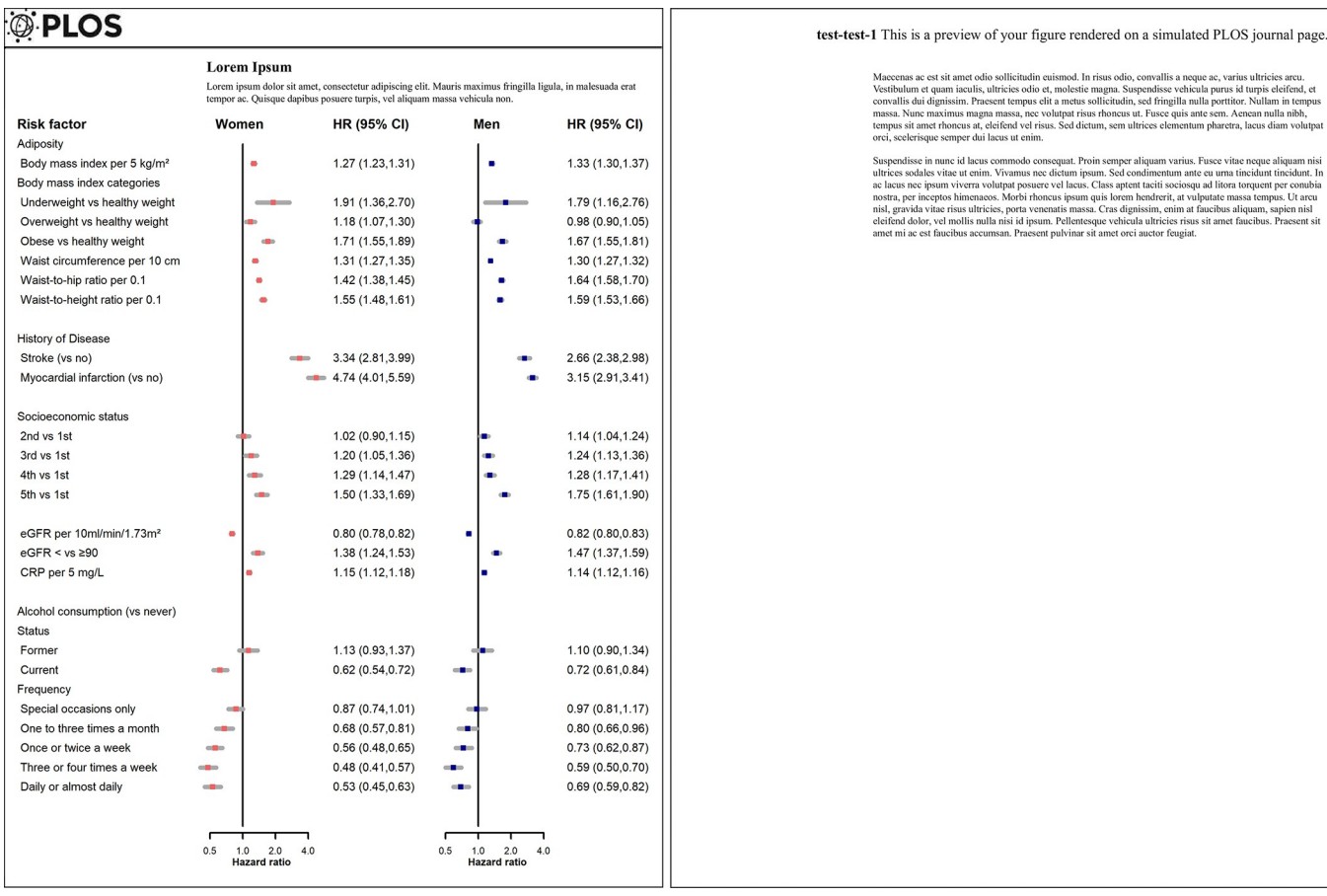

**Fig 3. Sex-specific multivariable-adjusted hazard ratios for non-traditional risk factors and PAD.** CRP denotes C-reactive protein, eGFRcys estimated Glomerular Filtration Rate calculated using cystatin C. Socioeconomic status was determined using the Townsend Deprivation Index and grouped into five groups based on the cut-offs for the UK national equal fifths, with the 1st group containing the least socially deprived and the 5th group the most deprived. Pink and blue squares represent hazard ratios for women and men, respectively. Horizontal lines indicate corresponding 95% confidence intervals.

### Inflammation

Per 1 mg/L greater CRP was associated with a higher risk of PAD in women 1.15 (1.12, 1.18) and in men 1.14 (1.12, 1.16).

### Alcohol consumption

For both sexes, compared with those who never drank alcohol, current alcohol drinkers had a lower risk of PAD. HRs were 0.62 (0.54, 0.72) in women and 0.72 (0.61, 0.84) in men. All self-reported frequencies of current alcohol consumption except for "drinking on special occasions only" were related to a lower risk of PAD in both sexes. Compared to those who never drank alcohol, the lower risk was more extreme for women than men who reported themselves as drinking "once or twice a week" RHR 0.76 (0.60, 0.96) and "daily or almost daily" RHR 0.77 (0.61. 0.98).

Results were broadly similar in the analyses adjusted for age only and in sensitivity analyses (S6–S9 Tables).

### Effect modification

There was no evidence that the women-to-men difference in association between any of the risk factors and PAD differed by age or smoking or diabetes status (S10–S12 Tables). Amongst

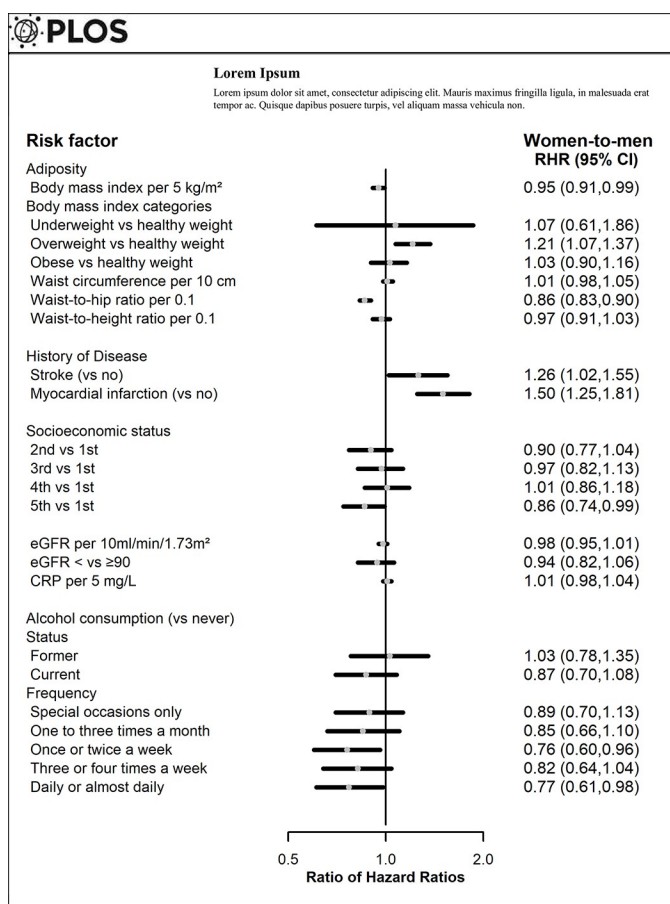

**Fig 4. Women-to-men ratio of hazard ratios for non-traditional risk factors and PAD.** CRP denotes C-reactive protein, eGFRcys estimated Glomerular Filtration Rate calculated using cystatin C. Socioeconomic status was determined using the Townsend Deprivation Index and grouped into five groups based on the cut-offs for the UK national equal fifths, with the 1st group containing the least socially deprived and the 5th group the most deprived. Squares represent women-to-men ratio of hazard ratios. Horizontal lines indicate corresponding 95% confidence intervals.

those of lower SES (below the national median), the excess risk conferred by a higher waist-to-hip ratio was higher in women than men, with the reverse true for those of higher SES (S13 Table). All other apparent differences are most likely due to chance (all $p$>0.001). There was no evidence that the women-to-men difference in association between PAD and blood pressure measures or lipid profiles differed by use of antihypertensive or lipid lowering medication, respectively (S14 and S15 Tables).

## Discussion

In the current study, among participants aged in a broad spectrum at baseline and followed up for over 10 years, the age-adjusted annual incidence of PAD hospitalisation or death was 79.6 and 185.3 per 100,000 population in women and men, respectively. This was comparable to the real-life age-adjusted data in Australia, where the rates in 2018–2019 (albeit being prevalent hospitalisation and not including death) were 77 and 152 per 100,000 population [19]. We assessed sex differences in a range of risk factors for PAD. Hypertension, type 1 and 2 diabetes, lower SES, lower kidney function, and higher CRP were each associated with a higher risk of PAD, similarly in both sexes. Being a current smoker, or having prior stroke or MI conferred 18%, 26%, or 50% higher excess risk of PAD, respectively, in women. Per 1 mmol/L higher

HDL-C was associated with lower risk of PAD, which was 34% greater in women than in men. Additionally, compared with those with HDL-C levels in the range of 40 to 60mg/dL, levels of ≤40 mg/dL was related to higher excess risk in women than men, and levels of >80 mg/dL was related to lower risk of PAD in women but higher risk in men.

In earlier studies in the UK Biobank, a higher relative risk of stroke [15] and MI [14] in women compared to men was found across all American Heart Association hypertension stages. This may be explained by a recent finding, where the brachial cuff measures underestimated invasive aortic SBP estimation, to a greater degree in women compared to men (mean differences: -6.5 mmHg for women versus -0.3 for men) [22]. However, a higher excess risk of PAD in women was not observed in any hypertension stages in our study. We found that higher SBP (per 10 mmHg) was related to 2% greater excess risk of PAD in women than in men. Yet, while PAD risk was log-linearly higher with higher SBP starting from 140 mmHg in men, such relationship was observed in all values of SBP in women. Therefore, whether and how blood pressure and hypertension impact differently on the risk of PAD in women and men is uncertain. There are also null findings from another large UK cohort registered in primary care, the strengths of the relationship between SBP and PAD were similar in both sexes (HR 1.63 (95% CI 1.59, 1.68) in women and 1.62 (95% CI 1.58, 1.66) in men) [20].

The dose-response relationship between number of cigarettes smoked and PAD, and the attenuation of risk with quitting, are in line with previous studies [21]. Women current smokers had a higher excess risk of PAD, possibly due to higher amount of toxic chemicals from same number of cigarettes they get than men [22]. This greater intake of toxic chemicals in women notably overweighted the extra harm caused by greater intensity and duration of exposure in men than women current smokers (32.1 versus 25.9 pack-years). Yet, in another study, the age-adjusted HR for current smoking (than never) and PAD was stronger in men 5.72 (5.24, 6.24) than in women 4.17 (3.68, 4.73) [23]. The benefits of smoking cessation in reducing CVD risk factors [24], or CVD [25] increased as the time since quitting increased. In this study, men and women who quitted had similar duration since quitting (averages of 19.2 versus 18.1 years). Thus, the greater relative risk of PAD in men former smokers might be explained by the fact that they had greater averaged number of cigarettes smoked per day along with similar durations between starting and quitting compared to women counterparts [26].

We found diabetes to be associated with a higher risk of PAD in both sexes, but no evidence of a sex disparity. This is consistent with a previous evidence synthesis, where the pooled adjusted relative risks (RRs) for incident PAD associated with diabetes, compared to no diabetes, were similar in women and in men, with the women-to-men ratio of RRs being 1.05 (0.90, 1.22) [27]. Furthermore, consistent with findings for the outcomes of MI and stroke [14, 15], in both sexes type 1 diabetes was more strongly associated with the risk of PAD than type 2 diabetes.

Higher TC and LDL-C were significant contributors to PAD [17]. Recently, it was found that the importance of cholesterol in PAD was largely supplanted by inflammation, diabetes, and smoking [28]. Accordingly, in terms of cholesterol, we only observed higher HDL-C being associated with a lower risk of PAD in both sexes, to a greater extent in women. Similar results were found in a Scottish population, where per 1 mmol/L higher HDL-C was related to a lower risk of PAD, age-adjusted HR 0.73 (0.61, 0.87) in women and 0.98 (0.87, 1.10) in men [28], corresponding to women-to-men RHR of 0.74 (0.60, 0.92). The greater excess risk in women than in men associated with low-level HDL-C and the sex-specific and opposite directions of associations between very high-level HDL-C and PAD are both consistent with a previous UK Biobank analysis [29].

Body adiposity was associated with the risk of PAD in both sexes. Similar associations of BMI and being overweight or obese in women and men on risk of incident CAD have been

suggested in an evidence synthesis [30]. Our findings were not in agreement with a previous study which found that the lowest risk of PAD was among those who were overweight rather than healthy weight, in both women and men [31]. Yet, in both studies, BMI categories affected, or tended to affect, the risk of PAD more strongly in women than men. The effect modification by SES on sex disparity in the association between waist-to-hip ratio and PAD, are likely to have been driven by our finding that waist-to-hip ratio was more strongly related to PAD risk in women in low compared to high SES ($p<0.001$). Consistently, women of low SES have been found to be more susceptible to obesity [32]. However, it is uncertain why this was not the case in other adiposity measures.

Prior stroke and MI were stronger risk factors for PAD in women than in men in our analyses. This potentially represents women's specific natural history of PAD, from onset to complications. Women with PAD have appeared not to present with typical symptoms [9]. Yet, low ABI (regardless of symptoms) was related to a higher risk of stroke and MI [33]. Among those who present to health services with a stroke or MI, a greater incidental detection of PAD might have occurred in women than men. Consequently, in women, PAD progression is more likely to present in the order of 1) asymptomatic low ABI, 2) complications of stroke and MI, and 3) PAD resulting in hospitalisation.

Better kidney function was related to a lower risk of PAD in both sexes. The null finding of sex disparity in the relationship of kidney function and PAD was in contrast to findings from a previous cohort, where women with chronic kidney disease have a higher PAD risk compared with men, especially before 70 years old [34]. As regards to inflammation, a higher CRP value was associated with PAD similarly in both sexes. Consistently, in the Scottish population, high-sensitivity CRP was related to higher risk of PAD similarly in women and men, multivariable adjusted HR 1.57 (1.35; 1.82) and 1.41 (1.23; 1.62), respectively [28]. Currently, while an overall harmful effect of alcohol on cardiometabolic health at all levels of alcohol use is suggested in some studies [35, 36], others also found that moderate alcohol drinkers had lower risk of incident PAD or cardiovascular outcomes compared to those who never drank, drank occasionally, or current abstainers [37–39]. Data used for alcohol consumption in the current study lack detail, such as the beverage type and quantity of units consumed. A possible explanation for our results might be that many current drinkers in this cohort were drinking in moderation (e.g., average weekly equivalent 7.7 glasses of wine [35]), especially those who drank three or four times a week.

To our knowledge, this study is the first to evaluate sex differences in the associations between a range of risk factors and PAD in a large prospective cohort of general population using standardised methodology. There are several limitations worth noting. First, most UK Biobank participants are White, and are more likely to live in less socioeconomically deprived areas, and be healthier, than the general population [40]. Second, self-reported baseline information on smoking and alcohol related variables might have introduced differential measurement errors by sexes. For instance, a recent Korean study found that cotinine (an objective measure of smoking inhalation) was higher in women than men at the same level of self-reported smoking [41], although this was not found in an earlier Scottish study [22]. Yet, although history of diabetes, stroke and MI, and use of antihypertensive and/or lipid lowering medication were also self-reported through touchscreen questionnaires, these were further confirmed in nurse-led interviews [42]. All the other variables were measured by the study team. Third, our assessment of PAD was based on hospitalisation or death. However, individuals with asymptomatic PAD (more common in women [9]) might not present to the healthcare system, and those who were on conservative management not requiring a procedure would not need hospitalisation. As a result, it is likely that we have missed PAD cases, which would lead to an underestimate of the real disease rates. The reported rates therefore represent those

of PAD that is serious enough to cause hospitalisation or death. That said, PAD hospitalisations represent a substantial medical and financial burden for patients and healthcare system [43], and thus, are worth investigation. Finally, we note that multiple tests were conducted and the related concern of false positives [44]. Thus, the results should be interpreted in the light of the effect sizes and clinical relevance.

## Conclusions

We compared the effects of risk factors for PAD hospitalisation or death by sex in the same study population on an equal basis and examined in subpopulations. Smoking cessation is important, especially for women. Our findings also suggest that clinicians should be vigilant when their female patients have a history of stroke or MI. The stronger associations of a higher waist-to-hip ratio with higher PAD risk in women may be amplified, when they are socioeconomically deprived.

Risk factor identification and management are clinically relevant. If the observed doubled risk of PAD documentation in men than women are due to sex differences in symptoms, identifying risk factors could increase the yield of ABI (not yet a routine screening test) in capturing asymptomatic PAD [45], and thus, benefit affected asymptomatic women. For premorbid prevention and people with PAD, strict control of risk factors is crucial [17]. Yet, more research is required to provide evidence to support or oppose sex-specific risk factor identification and management.

## Supporting information

**S1 Checklist. STROBE statement—checklist of items that should be included in reports of observational studies.**
(DOCX)

**S1 Fig. Sex-specific multivariable-adjusted hazard ratios for blood pressure measures with the risk of peripheral artery disease.**
(PDF)

**S2 Fig. Sex-specific multivariable-adjusted hazard ratios for cholesterol measures with the risk of peripheral artery disease.**
(PDF)

**S3 Fig. Sex-specific multivariable-adjusted hazard ratios for adiposity measures with the risk of peripheral artery disease.**
(PDF)

**S4 Fig. Sex-specific multivariable-adjusted hazard ratios for Townsend deprivation index with the risk of peripheral artery disease.**
(PDF)

**S5 Fig. Sex-specific multivariable-adjusted hazard ratios for other risk factors with the risk of peripheral artery disease.**
(PDF)

**S1 Table. Risk factors considered in the study and adjustments.**
(PDF)

**S2 Table. Diagnostic and procedure codes used in ascertaining peripheral artery disease.**
(PDF)

**S3 Table. Baseline characteristics of the UK Biobank participants excluded from the current analyses due to a previous hospital admission with peripheral artery disease.**
(PDF)

**S4 Table. Baseline characteristics of the UK Biobank participants included in the current analyses by PAD status.**
(PDF)

**S5 Table. Numbers and percentages of missing values for each variable.**
(PDF)

**S6 Table. Age-adjusted sex-specific hazard ratios and women-to-men ratio of hazard ratios for risk factors.**
(PDF)

**S7 Table. Sex-specific hazard ratios and women-to-men ratio of hazard ratios for risk factors in the sensitivity analysis.**
(PDF)

**S8 Table. Sex-specific hazard ratios and women-to-men ratio of hazard ratios for risk factors in an additional sensitivity analysis.**
(PDF)

**S9 Table. Sex-specific subdistribution hazard ratios and women-to-men ratio of subdistribution hazard ratios for risk factors.**
(PDF)

**S10 Table. Sex-specific multivariable-adjusted hazard ratios and women-to-men ratio of hazard ratios for risk factors by age group.**
(PDF)

**S11 Table. Sex-specific multivariable-adjusted hazard ratios and women-to-men ratio of hazard ratios for risk factors by smoking status.**
(PDF)

**S12 Table. Sex-specific multivariable-adjusted hazard ratios and women-to-men ratio of hazard ratios for risk factors by presence or absence of diabetes.**
(PDF)

**S13 Table. Sex-specific multivariable-adjusted hazard ratios and women-to-men ratio of hazard ratios for risk factors by socioeconomic status.**
(PDF)

**S14 Table. Sex-specific multivariable-adjusted hazard ratios and women-to-men ratio of hazard ratios for blood pressure by use of antihypertensive medication.**
(PDF)

**S15 Table. Sex-specific multivariable-adjusted hazard ratios and women-to-men ratio of hazard ratios for cholesterol measures by use of lipid lowering medication.**
(PDF)

## Acknowledgments

We thank the UK Biobank volunteers for their contribution.

## Author Contributions

**Conceptualization:** Mark Woodward.

**Data curation:** Katie Harris, Mark Woodward.

**Formal analysis:** Ying Xu, Katie Harris.

**Funding acquisition:** Ying Xu, Katie Harris, Cheryl Carcel, Mark Woodward.

**Investigation:** Ying Xu, Katie Harris, Anna Louise Pouncey, Cheryl Carcel, Gary Low, Sanne A. E. Peters, Mark Woodward.

**Methodology:** Ying Xu, Katie Harris, Anna Louise Pouncey, Cheryl Carcel, Gary Low, Sanne A. E. Peters, Mark Woodward.

**Supervision:** Mark Woodward.

**Validation:** Ying Xu, Katie Harris, Anna Louise Pouncey, Cheryl Carcel, Gary Low, Sanne A. E. Peters, Mark Woodward.

**Visualization:** Ying Xu, Katie Harris.

**Writing – original draft:** Ying Xu.

**Writing – review & editing:** Ying Xu, Katie Harris, Anna Louise Pouncey, Cheryl Carcel, Gary Low, Sanne A. E. Peters, Mark Woodward.

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
