## [Decision Letter · Decision Letter 0]

10 Jul 2023

PONE-D-23-13976Sex differences in risk factors for incident peripheral artery disease hospitalization or death: cohort study of UK Biobank participantsPLOS ONE

Dear Dr. Xu,

Thank you for submitting your manuscript to PLOS ONE. After careful consideration, we feel that it has merit but does not fully meet PLOS ONE’s publication criteria as it currently stands. Therefore, we invite you to submit a revised version of the manuscript that addresses the points raised during the review process.

We look forward to receiving your revised manuscript.

Kind regards,

Ahmed Arafa

Academic Editor

PLOS ONE

“I have read the journal's policy and the authors of this manuscript have the following competing interests: MW has done recent consultancy for Amgen and Freeline outside the submitted work; no support from any organization; for the submitted work; no other relationships or activities that could appear to have influenced the submitted work. All other authors have nothing to declare.”

Additional Editor Comments:

1- Alcohol consumption was shown in previous articles to be associated with PAD. It should be investigated in this article as well.

2- Recent UK Biobank reports showed higher risk of mortality and CVD among those with very high HDL-C (>80 mg/dL); thus, categorizing HDL-C should be considered.

Reviewers' comments:

Reviewer's Responses to Questions

**Comments to the Author**

1. Is the manuscript technically sound, and do the data support the conclusions?

Reviewer #1: Yes

Reviewer #2: Yes

2. Has the statistical analysis been performed appropriately and rigorously? 

Reviewer #1: Yes

Reviewer #2: Yes

3. Have the authors made all data underlying the findings in their manuscript fully available?

Reviewer #1: No

Reviewer #2: Yes

4. Is the manuscript presented in an intelligible fashion and written in standard English?

Reviewer #1: Yes

Reviewer #2: Yes

5. Review Comments to the Author

Reviewer #1: In a large cohort study of 500,207 UK Biobank participants without prior hospitalization of peripheral artery disease (PAD) at baseline, the authors reported that smoking and a history of stroke or MI were more strongly associated with a higher risk of PAD in women than men. HDL-C was more strongly associated with a lower risk of PAD in women than men. The strengths of this study include its large sample size, and the manuscript is generally well written and has great scientific merit.

Major Comments

1. In abstract lines 41-45, please state the comparator. For example, the smokers were compared to never smokers, not to non-current smokers.

2. In lines 115-116, the authors did the complete case analyses. How many were excluded because of missing covariates?

3. How did the authors deal with competing risk events such as other CVD or death in the survival analysis? This reviewer would recommend the additional analyses to account for the competing events.

4. Ethical approval and data availability of this study should be stated in the methods section.

Reviewer #2: This prospective cohort study, which included 500,207 UK Biobank participants, aimed to investigate the association between risk factors and the risk of peripheral artery disease, specifically focusing on the potential sex differences. The large number of participants is a significant strength, ensuring a robust and reliable conclusion. The data analysis conducted in this study is comprehensive, providing detailed insights and thorough examination of the collected information.

Major comments:

1. In the Introduction section, on line 53, the authors stated that "It is uncertain whether women or men exhibit a greater risk of PAD [1-3]." However, reference 1 has reported sex differences in PAD. According to the pooled odds ratio of 29 studies, the risk for peripheral artery disease is 0.74 (0.61–0.91) in males compared to females (Table 3, Risk factor 2: male sex).

Additionally, previous studies have reported differences in the incidence of peripheral artery disease between sexes. The sex differences are not novel to clinicians either. So, I think that the statement "It is uncertain" is not appropriate.

Hicks CW, Ding N, Kwak L, Ballew SH, Kalbaugh CA, Folsom AR, Heiss G, Coresh J, Black JH 3rd, Selvin E, Matsushita K. Risk of peripheral artery disease according to race and sex: The Atherosclerosis Risk in Communities (ARIC) study. Atherosclerosis. 2021 May;324:52-57. doi: 10.1016/j.atherosclerosis.2021.03.031.

Kalbaugh CA, Kucharska-Newton A, Wruck L, Lund JL, Selvin E, Matsushita K, Bengtson LGS, Heiss G, Loehr L. Peripheral Artery Disease Prevalence and Incidence Estimated From Both Outpatient and Inpatient Settings Among Medicare Fee-for-Service Beneficiaries in the Atherosclerosis Risk in Communities (ARIC) Study. J Am Heart Assoc. 2017 May 3;6(5):e003796. doi: 10.1161/JAHA.116.003796.

2. The inconsistent results reported in previous studies and this study in the prevalence and incidence of peripheral artery disease between sexes are a matter of concern. The authors discussed this issue in the second paragraph of the Discussion section. However, the discussion is vague and lacks specific hypotheses about the cause of this discrepancy.

3. In the Discussion section, on lines 249-250, the authors stated that "However, a higher excess risk of PAD in women was not observed in any hypertension stages in our study. ". I don't quite understand this sentence, because I found that the hazard ratio (95% CI) is 1.36 (1.18-1.56) for stage 2 hypertension versus normal blood pressure in female.

Besides, in the same paragraph, the authors discussed the differences in the dose-response relationship between sexes, and stated that "This reflects the controversy on whether blood pressure lowering target is “the lower, the better”. I don't quite understand why it reflects this controversy. The ideal blood pressure levels should be considered based on the specific patient's situation. For instance, in the older population or in patients with kidney disease or other serious underlying conditions, it is not necessarily true that “the lower, the better”. I do not feel that the authors' analysis reflects these backgrounds of patients.

Minor comments:

1. (mean (SD)) is missing for “Pack-years among current smokers” and “Year since quitting among former smokers” in Table 1.

2. I feel that the layout of the Tables is a little messy, and it is not easy to understand its content. For example, I can't tell which is the mean value (SD) and which is N (%). Some expressions use too many brackets, such as "Systolic blood pressure (mmHg) (mean (SD))". Authors may consider incorporating some formatting or using bold subheadings to make it easier to read. For instance, "American Heart Association Hypertension Categories." Alternatively, these issues could be addressed by the publishing team?

It is advisable to include more descriptive information in the supplemental tables. For instance, adding a description that clarifies the type of sensitivity analysis conducted in S5 Table and S6 Table.

3. Could the authors consider showing a forest plot for the ratios of hazard ratio for women versus men in Fig 1?

6. PLOS authors have the option to publish the peer review history of their article (what does this mean?). If published, this will include your full peer review and any attached files.

Reviewer #1: No

Reviewer #2: No

---

## [Author Response · Author response to Decision Letter 0]

29 Aug 2023

19 August 2023

The Academic Editor

RE: Consideration of publication of our manuscript (PONE-D-23-13976) - Sex differences in risk factors for incident peripheral artery disease hospitalisation or death: cohort study of UK Biobank participants

Dear Dr Ahmed Arafa

We appreciate the opportunity to address the Reviewers’ comments and have revised our manuscript. Below, please find item-by-item responses to your and Reviewers’ comments. All the line numbers mentioned in this document refer to the document with tracked changes.

Yours sincerely

Ying Xu

Research Fellow at The George Institute for Global Health

Additional Editor Comments:

1. Alcohol consumption was shown in previous articles to be associated with PAD. It should be investigated in this article as well.

Response: Thanks for this suggestion. We have added variables of alcohol drinker status and frequency of alcohol consumption.

• In the Methods, S1 Table “Alcohol drinking status was self-reported as never, former, or current alcohol drinker. Frequency of alcohol consumption among current drinkers was self-reported as: special occasions only, one to three times a month, once or twice a week, three or four times a week, and daily or almost daily.”

• In the Results and under the subheading of “Alcohol consumption”, “For both sexes, compared with those who never drank alcohol, current alcohol drinkers had a lower risk of PAD. HRs were 0.62 (0.54, 0.72) in women and 0.72 (0.61, 0.84) in men. All self-reported frequencies of current alcohol consumption except for “drinking on special occasions only” were related to a lower risk of PAD in both sexes. Compared to those who never drank alcohol, the lower risk was more extreme for women than men who reported themselves as drinking “once or twice a week” RHR 0.76 (0.60, 0.96) and “daily or almost daily” RHR 0.77 (0.61. 0.98).” (lines 252 to 258)

• Figs 3-4, Tables 1-2, and S7-S13 Tables have also been updated, adding information and analyses on alcohol consumption.

• In the Discussion, “Currently, while an overall harmful effect of alcohol on cardiometabolic health at all levels of alcohol use is suggested in some studies [35, 36], others also found that moderate alcohol drinkers had lower risk of incident PAD or cardiovascular outcomes compared to those who never drank, drank occasionally, or current abstainers [37-39]. Data used for alcohol consumption in the current study lack detail, such as the beverage type and quantity of units consumed. A possible explanation for our results might be that many current drinkers in this cohort were drinking in moderation (e.g., average weekly equivalent 7.7 glasses of wine [35]), especially those who drank three or four times a week.” (lines 379 to 387)

• We also noted that information on alcohol consumption is self-reported and might introduce measurement bias. This limitation reads, “Second, self-reported baseline information on smoking and alcohol related variables might have introduced differential measurement errors by sexes.” (lines 392 to 394)

2. Recent UK Biobank reports showed higher risk of mortality and CVD among those with very high HDL-C (>80 mg/dL); thus, categorizing HDL-C should be considered.

Response: Thanks for this interesting suggestion. We have now added results for categorized HDL-C.

• In the Abstract, “Compared to HDL-C at 40 to 60 mg/dL, the lowest level of HDL-C (≤40 mg/dL) was related to greater excess risk in women, RHR 1.20 (1.02, 1.41), whereas the highest level of HDL-C (>80 mg/dL) was associated with lower risk of PAD in women, but higher risk in men, RHR 0.50 (0.38, 0.65).” (lines 49 to 52)

• In the Methods, S1 Table “Five groups were defined based on high-density lipoprotein cholesterols levels: 30, 40, 60, 80 mg/dL, equivalent to 0.78, 1.03, 1.55, and 2.07 mmol/L. Participants in the first two groups were combined into one group in the Cox and Fine and Gray models, due to small sample sizes.” 

• In the Results, “Compared to HDL-C at 40 to 60 mg/dL, the lowest level of HDL-C (≤40 mg/dL) was related to higher excess risk in women than men RHR 1.20 (1.02, 1.41); whereas the highest level of HDL-C (>80 mg/dL) was associated with lower risk of PAD in women HR 0.79 (0.66, 0.94), but higher risk in men HR 1.58 (1.30, 1.93), women-to-men RHR 0.50 (0.38, 0.65).” (lines 197 to 200)

• Tables 1-2, S7-S13 Tables, and S15 Table have also been updated, adding HDL-C categories.

• In the first paragraph of the Discussion, “Additionally, compared with those with HDL-C levels in the range of 40 to 60mg/dL, levels of ≤40 mg/dL was related to higher excess risk in women than men, and levels of >80 mg/dL was related to lower risk of PAD in women but higher risk in men.” (lines 282 to 284)

• In the Discussion, “The greater excess risk in women than in men associated with low-level HDL-C and the sex-specific and opposite directions of associations between very high-level HDL-C and PAD are both consistent with a previous UK Biobank analysis [29].” (lines 350 to 353)

Reviewers' comments:

Reviewer #1: In a large cohort study of 500,207 UK Biobank participants without prior hospitalization of peripheral artery disease (PAD) at baseline, the authors reported that smoking and a history of stroke or MI were more strongly associated with a higher risk of PAD in women than men. HDL-C was more strongly associated with a lower risk of PAD in women than men. The strengths of this study include its large sample size, and the manuscript is generally well written and has great scientific merit.

Major Comments

1. In abstract lines 41-45, please state the comparator. For example, the smokers were compared to never smokers, not to non-current smokers.

Response: Comparators have been added as suggested. The sentence reads, “Compared with men, women who were smokers or had a history of stroke or MI had a greater excess risk of PAD (relative to those who never smoked or had no history of stroke or MI): RHR 1.18 (95%CI 1.04, 1.34), 1.26 (1.02, 1.55), and 1.50 (1.25, 1.81), respectively.” (lines 44 to 47)

2. In lines 115-116, the authors did the complete case analyses. How many were excluded because of missing covariates?

Response: We have added S5 Table: Numbers and percentages of missing values for each variable, and mentioned in the text that “Percentages of missing for each variable were <7% (S5 Table), except for high-density lipoprotein cholesterol (HDL-C, 14.4%).” (lines 137 to 139)

3. How did the authors deal with competing risk events such as other CVD or death in the survival analysis? This reviewer would recommend the additional analyses to account for the competing events.

Response: We added Fine-Gray regression models. In the Methods, “In a further sensitivity analysis, we used Fine-Gray regression models and examined the relationships between risk factors and incident PAD documentation considering a competing event, i.e., death due to reasons other than PAD. Participants who did not experience PAD hospitalisation but experienced the competing event were treated as being censored at infinity to indicate that they would never experience PAD [18].” (lines 126 to 131) The relevant results were reported in S9 Table and in the text, “Results were broadly similar in the analyses adjusted for age only and in sensitivity analyses (S6-9 Tables).” (lines 259 to 260) In our opinion, other CVD should not be treated in the same way, because experiencing other CVD does not preclude a subsequent PAD.

4. Ethical approval and data availability of this study should be stated in the methods section.

Response: We added in the Methods that “The UK Biobank obtained ethics approval from the National Health Service`s National Research Ethics Service Committee (ref 21/NW/0157). This research was conducted using UK Biobank resource (application 74018) approved by the access subcommittee.” (lines 86 to 89)

Reviewer #2: This prospective cohort study, which included 500,207 UK Biobank participants, aimed to investigate the association between risk factors and the risk of peripheral artery disease, specifically focusing on the potential sex differences. The large number of participants is a significant strength, ensuring a robust and reliable conclusion. The data analysis conducted in this study is comprehensive, providing detailed insights and thorough examination of the collected information.

Major comments:

1. In the Introduction section, on line 53, the authors stated that "It is uncertain whether women or men exhibit a greater risk of PAD [1-3]." However, reference 1 has reported sex differences in PAD. According to the pooled odds ratio of 29 studies, the risk for peripheral artery disease is 0.74 (0.61–0.91) in males compared to females (Table 3, Risk factor 2: male sex).

Additionally, previous studies have reported differences in the incidence of peripheral artery disease between sexes. The sex differences are not novel to clinicians either. So, I think that the statement "It is uncertain" is not appropriate.

Hicks CW, Ding N, Kwak L, Ballew SH, Kalbaugh CA, Folsom AR, Heiss G, Coresh J, Black JH 3rd, Selvin E, Matsushita K. Risk of peripheral artery disease according to race and sex: The Atherosclerosis Risk in Communities (ARIC) study. Atherosclerosis. 2021 May;324:52-57. doi: 10.1016/j.atherosclerosis.2021.03.031.

Kalbaugh CA, Kucharska-Newton A, Wruck L, Lund JL, Selvin E, Matsushita K, Bengtson LGS, Heiss G, Loehr L. Peripheral Artery Disease Prevalence and Incidence Estimated From Both Outpatient and Inpatient Settings Among Medicare Fee-for-Service Beneficiaries in the Atherosclerosis Risk in Communities (ARIC) Study. J Am Heart Assoc. 2017 May 3;6(5):e003796. doi: 10.1161/JAHA.116.003796.

Response: Thank-you for these helpful references. We have rewritten this sentence and provided more details. “Men have higher age-standardised cardiovascular disease (CVD) rates [2], and so the significance of CVD, including PAD, in women has been under-recognised and under-investigated [3, 4]. The prevalence and incidence of PAD are similar in women and men with overlapping confidence intervals (CIs), and there is no clear agreement regarding whether women or men exhibit a greater PAD risk [1, 5-8]. There is, however, evidence to suggest sex disparities in PAD presentation, potentially leading to worse prognosis in women.” (lines 60 to 66).

2. The inconsistent results reported in previous studies and this study in the prevalence and incidence of peripheral artery disease between sexes are a matter of concern. The authors discussed this issue in the second paragraph of the Discussion section. However, the discussion is vague and lacks specific hypotheses about the cause of this discrepancy.

Response: To reduce confusion, we have removed this paragraph. In the first paragraph of the Discussion, we had provided comparisons from real life data, which are the most comparable to the current study in terms of the methodology. Otherwise, the rates would arguably vary with variations in PAD assessment methodologies, length of follow-up, and study populations, etc.

3. In the Discussion section, on lines 249-250, the authors stated that "However, a higher excess risk of PAD in women was not observed in any hypertension stages in our study. ". I don't quite understand this sentence, because I found that the hazard ratio (95% CI) is 1.36 (1.18-1.56) for stage 2 hypertension versus normal blood pressure in female.

Besides, in the same paragraph, the authors discussed the differences in the dose-response relationship between sexes, and stated that "This reflects the controversy on whether blood pressure lowering target is “the lower, the better”. I don't quite understand why it reflects this controversy. The ideal blood pressure levels should be considered based on the specific patient's situation. For instance, in the older population or in patients with kidney disease or other serious underlying conditions, it is not necessarily true that “the lower, the better”. I do not feel that the authors' analysis reflects these backgrounds of patients.

Response: Individuals with stage 2 hypertension had higher risk of PAD compared to individuals with normal blood pressure, which is true in both women and men in the analyses. Yet, the risk related to stage 2 hypertension is similar in women and men, reflected by RHR of 1.18 (0.99 to 1.42), which means there are no excess risk of PAD in women. We have removed the mentioning of the controversy, which truly is irrelevant (lines 310 to 316).

Minor comments:

1. (mean (SD)) is missing for “Pack-years among current smokers” and “Year since quitting among former smokers” in Table 1.

Response: Thanks for pointing out this error. “(mean (SD))” has been added (Table 1, S3 Table, and S4 Table).

2. I feel that the layout of the Tables is a little messy, and it is not easy to understand its content. For example, I can't tell which is the mean value (SD) and which is N (%). Some expressions use too many brackets, such as "Systolic blood pressure (mmHg) (mean (SD))". Authors may consider incorporating some formatting or using bold subheadings to make it easier to read. For instance, "American Heart Association Hypertension Categories." Alternatively, these issues could be addressed by the publishing team?

It is advisable to include more descriptive information in the supplemental tables. For instance, adding a description that clarifies the type of sensitivity analysis conducted in S5 Table and S6 Table.

Response: Tables have been tidied up. More details have been provided for S7 and S8 Tables (originally S5 and S6 Tables). 

The first footnote of S7 Table reads, “A record of peripheral artery disease could be identified purely based on a procedure that might be conducted for aneurysms (S1 Table). In this rare circumstance, those with a diagnosis of aneurysms before or at the time of the procedure were not counted as peripheral artery disease.”

The first footnote of S8 Table reads, “In this sensitivity analysis, only diagnostic codes for peripheral artery disease (S1 Table) were used.”

3. Could the authors consider showing a forest plot for the ratios of hazard ratio for women versus men in Fig 1?

Response: The ratios of hazard ratios were plotted, but due to the requirement on the width of figures, we must present them in four figures (Figs 1-4).

---

## [Editor Report · Decision Letter 1]

12 Sep 2023

Sex differences in risk factors for incident peripheral artery disease hospitalisation or death: cohort study of UK Biobank participants

PONE-D-23-13976R1

Dear Dr. Ying Xu,

We’re pleased to inform you that your manuscript has been judged scientifically suitable for publication and will be formally accepted for publication once it meets all outstanding technical requirements.

Kind regards,

Ahmed Arafa

Academic Editor

PLOS ONE
---

## [Editor Report · Acceptance letter]

18 Sep 2023

PONE-D-23-13976R1 

Sex differences in risk factors for incident peripheral artery disease hospitalisation or death: cohort study of UK Biobank participants 

Dear Dr. Xu:

I'm pleased to inform you that your manuscript has been deemed suitable for publication in PLOS ONE. Congratulations! Your manuscript is now with our production department. 

Kind regards, 

on behalf of

Dr. Ahmed Arafa 

Academic Editor

PLOS ONE